# Tectomer-Mediated Optical Nanosensors for Tyramine Determination

**DOI:** 10.3390/s23052524

**Published:** 2023-02-24

**Authors:** Mario Domínguez, Sofía Oliver, Rosa Garriga, Edgar Muñoz, Vicente L. Cebolla, Susana de Marcos, Javier Galbán

**Affiliations:** 1Nanosensors and Bioanalytical Systems (N&SB), Analytical Chemistry Department, Faculty of Sciences, Instituto de Nanociencia y Materiales de Aragón (INMA University of Zaragoza-CSIC), 50009 Zaragoza, Spain; 2Departamento de Química-Física, University of Zaragoza, 50009 Zaragoza, Spain; 3Instituto de Carboquímica ICB-CSIC, 50018 Zaragoza, Spain

**Keywords:** tyramine, biogenic amines, gold nanoparticles, RGB coordinates, optical sensors, food quality control, colorimetric sensors, tectomers

## Abstract

The development of optical sensors for in situ testing has become of great interest in the rapid diagnostics industry. We report here the development of simple, low-cost optical nanosensors for the semi-quantitative detection or naked-eye detection of tyramine (a biogenic amine whose production is commonly associated with food spoilage) when coupled to Au(III)/tectomer films deposited on polylactic acid (PLA) supports. Tectomers are two-dimensional oligoglycine self-assemblies, whose terminal amino groups enable both the immobilization of Au(III) and its adhesion to PLA. Upon exposure to tyramine, a non-enzymatic redox reaction takes place in which Au(III) in the tectomer matrix is reduced by tyramine to gold nanoparticles, whose reddish-purple color depends on the tyramine concentration and can be identified by measuring the RGB coordinates (Red–Green–Blue coordinates) using a smartphone color recognition app. Moreover, a more accurate quantification of tyramine in the range from 0.048 to 10 μM could be performed by measuring the reflectance of the sensing layers and the absorbance of the characteristic 550 nm plasmon band of the gold nanoparticles. The relative standard deviation (RSD) of the method was 4.2% (*n* = 5) with a limit of detection (LOD) of 0.014 μM. A remarkable selectivity was achieved for tyramine detection in the presence of other biogenic amines, especially histamine. This methodology, based on the optical properties of Au(III)/tectomer hybrid coatings, is promising for its application in food quality control and smart food packaging.

## 1. Introduction

The development of new sensors for the fast and cost-effective detection of toxic compounds is of great interest to the food industry [1,2,3,4,5]. In this sense, optical sensors for in situ diagnostics have become an alternative to instrumental separation techniques such as high-resolution thin-layer chromatography [6], capillary electrophoresis [7], and especially high performance liquid chromatography (HPLC) [8], which provide accurate and reliable results but are time-consuming and require more specialized personal. The remarkable optical properties of nanomaterials such as nanoparticles [9] or nanoclusters [10] of noble (Au or Pt) or semi-noble (Ag or Cu) metals, quantum dots [11], or nanoparticles with upconversion properties [12] are particularly attractive for the design of these novel sensors. Some of these zero-dimensional (0D) nanomaterials can even replace organic chromophores and fluorophores [13] due to their larger Stokes displacements, longer fluorescence lifetimes, narrower fluorescence spectra, and higher photostability. Furthermore, the analytical performance of these 0D material-based optical nanosensors can be significantly enhanced if the measured optical signal is related to the formation of nanoparticles during specific chemical reactions [14].

Amino-terminated oligoglycine peptides consist of two to four antennae of typically four to seven glycine units connected by alkyl linkers (Figure 1). These oligopeptides self-assemble through polyglycine II cooperative hydrogen bonding, resulting in the formation of two-dimensional (2D) structures denoted as tectomers [15,16], hundreds of nanometers to several micrometers in size, and a few nanometers thick (~5.6 nm and 4.5 nm for biantennary [Gly_4_-NH-CH_2_]_2_ C_8_H_16_ and tetraantennary [Gly_7_-NH-CH_2_]_4_C tectomer platelets, respectively, [17,18]). The 2D structure and surface amino groups of tectomers allow them to interact with a range of molecules and nanomaterials including metal nanoparticles as well as to different types of substrates. These adhesive properties of tectomer platelets are attractive for applications in functional coatings and sensing devices [18,19,20].

Tyramine is a biogenic amine (BA) formed by the decarboxylation of the amino acid tyrosine. Tyramine acts in the human body as a neuromodulator, affecting both the cardiovascular and immune systems [21,22]. Tyramine production is also associated with aged or spoiled foods. Symptoms of tyramine poisoning include nausea, migraines, hypertension, and respiratory problems [23]. Oxidative deamination to the corresponding aldehyde catalyzed by tyramine oxidase (TAO) is the physiological detoxification mechanism for tyramine. However, there are individuals who are intolerant to tyramine and other BAs, either because their bodies are unable to produce the corresponding amine oxidases, or because of enzyme inhibition due to drug ingestion. Therefore, there is a need for the rapid detection of BA in the food industry and for medical diagnosis and treatment [24,25]. Tyramine concentrations in representative foods considered to be risky for the above-mentioned individuals are listed in Table 1.

Previous work has reported on tyramine colorimetric sensors based on the redox oxidative deamination catalyzed by TAO and simultaneous determination by the reduction of Au(III) to Au^0^ in the form of gold nanoclusters (reddish-purple gold nanoparticles (AuNPs)) [14] or the HRP/TMB indicating reaction [26]. We here report on the development of novel non-enzymatic colorimetric nanosensors for tyramine detection based on Au(III)/tectomer sensing layers. Tectomer coatings on polylactic acid (PLA) supports act as matrices for tetrachloroauric (III) acid immobilization, enabling the non-enzymatic reduction of Au(III) to Au^0^, and hence gold clustering and subsequent gold nanoparticle (AuNP) formation upon the exposure to tyramine, resulting in a change of color in the Au(III)/tectomer layer as a function of tyramine concentration. Selectivity toward other interfering biogenic amines was achieved. The tectomer-mediated optical nanosensors presented here are promising for smart food packaging and biomedical applications. This new methodology allows for the formation of the nanomaterial in a single step and has many other advantages such as avoiding the use of dyes and not requiring the use of enzymes.

In this work, we combine the use of tectomers to immobilize optical biosensors and the in situ generation of nanoparticles to develop a simple method for the determination of tyramine (p-hydroxyphenethylamine).

## 2. Materials and Methods

### 2.1. Reagents and Solutions

All chemicals were used without further purification: Na_2_HPO_4_ ≥ 99% (Panreac (Barcelona, Spain) 131679.1211), Na_2_CO_3_ ≥ 99.5% (Sigma (Saint Louis, MO, USA) EC 207-838-8), CH_3_-COONa ~ 100% (VWR Chemicals (Darmstadt, Germany) 27648.294), TCA (Scharlab (Spain) NS15390100), tetrachloroauric (III) acid hydrate 99.995% (HAuCl_4_·3H_2_O, Stream Chemicals (Newburyport, MA, USA) 79-0500), tyramine hydrochloride ≥ 98% (Sigma (MO, USA) T2879), cadaverine dihydrochloride ~98% (Sigma (MO, USA) C85619), putrescine dihydrochloride ≥ 98% (Sigma (MO, USA) P7505), histamine dihydrochloride ≥ 99.0% (Sigma (MO, USA) 53300), biantennary oligoglycine [Gly_4_-NHCH_2_]_2_ C_8_H_16_·2HCl 95% (PlasmaChem GmbH (Berlin, Germany), Cat. No.: PL-TEC-2).

### 2.2. Equipment

Spectroscopic measurements were performed using an Agilent 8453A photodiode UV–Vis spectrophotometer, a SPECORD^R^ 210 Plus UV–Vis molecular absorption spectrophotometer, and a Cary Eclipse Fluorescence Spectrophotometer (Agilent Technologies (Santa Clara, CA, USA)) equipped with a 96-well microplate reader accessory. RGB measurements were carried out with a Huawei P30 mobile phone camera and the ColorGrab™ v. 3.6.1 color recognition app (Loomatix©).

### 2.3. Sensing Layer Fabrication

Au(III)/tectomer sensing layers were prepared from 20 µL of a premix of tetrachloroauric(III) acid and biantennary oligoglycine 3.0 × 10^−3^ M solutions that were drop-cast at room temperature on PLA supports (Biopack), 250 μm thick and 10 mm in diameter (Figure 2). The optimal experimental conditions were fixed at a Au(III)/tectomer 1:1 molar ratio and pH 6.0 phosphate buffer, as described below. After letting the films dry for one day, the films were immersed in 600 µL of Milli-Q water for 30 min to remove non-tightly assembled tectomer platelets.

To assess the ability of the tectomer to immobilize Au(III), the Au(III)/tectomer layers fabricated on PLA supports were placed inside a well-plate and 300 µL of pH 2.0 buffer solution was added, providing an acidic environment that triggers tectomer disassembly [17]. After 180 min, the resulting supernatant was analyzed using a method based on AuBr_4_^−^ complex formation and UV–Vis spectra measurements [27] as well as a standard ICP-OES (Thermo Scientific iCAP PRO XP Duo) method (see Appendix A.

### 2.4. Colorimetric Determination of Tyramine from RGB Coordinates

The smartphone was placed in a fixed position with a laboratory stand at 30 cm above the Au(III)/tectomer sensing layer on the PLA support and placed inside a well-plate, under constant lighting conditions. The distance between the smartphone and the sensing layers was optimized in the previous work of the research group [28]. Then, 300 µL of the tyramine solutions in pH 6.0 phosphate buffer was added. After 180 min, the RGB coordinates of the sheet were recorded with the ColorGrab™ app on the smartphone.

From these coordinates, R offers the higher sensitivity when measuring the reddish-purple color of the sheets, which was obtained upon the addition of tyramine due to the formation of AuNPs. However, in order to obtain measurements independent of the smartphone and the lighting conditions, the parameter used for the calibration curve was *R_r_*:(1)$$R_{r}=\frac{\left(R_{0}-R\right)}{R_{0}}\;$$

*R*_0_ is the value of the coordinate *R* corresponding to the blank (pH 6.0 phosphate buffer), that is, in the absence of the analyte, and *R* is the value of the coordinate upon the addition of tyramine in pH 6.0 phosphate buffer solution after 180 min.

### 2.5. Colorimetric Determination of Tyramine from Absorbance Measurements

Tyramine quantification can more accurately be measured from the reflectance measurements collected using the fluorescence spectrophotometer. Au(III)/tectomer sensing layers on PLA supports were placed in the microplate reader and the fluorescence intensity was acquired with the “synchronous scanning” function (which consists of exciting the sample at the same wavelength at which the emitted light is acquired) in the 400–800 nm range; the spectrum obtained corresponded to the *I*_0,*λ*_ value of each wavelength. After the addition of 300 µL of the corresponding tyramine solution, the fluorescence synchronous scanning was performed again and the spectrum obtained corresponded to *I_t,λ_*. The complete absorption spectrum was then calculated as:(2)$$Abs_{\lambda }=-log\frac{I_{t,\lambda }}{I_{0,\lambda }}\;$$

## 3. Results and Discussion

### 3.1. Tectomer-Mediated Au(III) Immobilization on PLA Supports

Au(III) was effectively immobilized by means of a tectomer on PLA supports, as discussed in Appendix A. The adhesion of the sensing layers on the PLA supports was favored by the hydrogen bond formation between the surface amino groups of the tectomers and the carboxyl groups of PLA.

### 3.2. Sensing Based on RGB Measurements

The reducing properties of tyramine leads to a non-enzymatic redox reaction in which Au(III) reduction occurs in the tectomer matrix, resulting in the formation of AuNPs. Thus, Au(III)/tectomer coatings on the PLA supports were placed inside a well-plate and exposed to different tyramine solutions (0, 3, 10, 30, and 100 µM). After a reaction time of 180 min, the color of the sensing layers changed as a function of the tyramine concentration, as shown in Figure 3. The color of the sensor layer changed from reddish to purple as the tyramine concentration increased due to the increasing size of the resulting AuNPs.

The RGB coordinates were measured to quantify the effect of tyramine oxidation leading to AuNP formation; specifically, the R coordinate was plotted as a function of the tyramine concentration due to its higher sensitivity to the color of the AuNPs. The optimal experimental conditions for sensor fabrication were found by premixing Au(III) and biantennary oligoglycine solutions in a molar ratio of 1:1 in pH 6.0 phosphate buffer (see Appendix A.

The analytical characterization of the proposed sensor pursued two goals, namely, to define an in situ screening method for tyramine from the RGB coordinates and to define a quantitative determination procedure from the reflectance measurements collected with a fluorimeter. With regard to the in situ screening of tyramine, our research group developed a mathematical model [26] for the measurement of RGB coordinates in solid supports in order to obtain more reproducible data and to compare them with those measured using other smartphones. According to this model, the following equation relates the *R* coordinate to the molar concentration of the absorbing species in the sensing layer:(3)$$R=A\left(E_{0,R}-E_{1,R}c+E_{2,R}c^{2}\right)$$
where *A* is a constant that considers parameters such as camera design, measurement angle, light-to-voltage conversion, and analog-to-digital conversion. On the other hand, R is the coordinate of the color that is measured, and *E*_0*,R*_, *E*_1*,R*_ and *E*_2*,R*_ are given by the following equations:(4)$$E_{0,R}=\underset{\lambda }{\sum }\left[I_{\lambda }P_{R,\lambda }\left(\frac{s_{\lambda }L}{1+s_{\lambda }L}\right)\right]$$
(5)$$E_{1,R}=2.3\underset{\lambda }{\sum }\left[I_{\lambda }P_{R,\lambda }\varepsilon_{\lambda }{\left(\frac{s_{\lambda }L}{1+s_{\lambda }L}\right)}^{2}\left(\frac{3+2s_{\lambda }L\;}{3s_{\lambda }L}\right)\right]$$
(6)$$E_{2,R}=5.3\underset{\lambda }{\sum }\left[I_{\lambda }P_{R,\lambda }\varepsilon_{\lambda }^{2}{\left(\frac{s_{\lambda }L}{1+s_{\lambda }L}\right)}^{3}\left(\frac{30+45s_{\lambda }L+24\;{\left(s_{\lambda }L\right)}^{2}+4\;{\left(s_{\lambda }L\right)}^{3}\;}{45s_{\lambda }^{2}}\right)\right]$$

In these equations, *I_λ_* and *P_R,λ_* are the spectral power of the illumination source and the spectral sensitivity of the camera (defined as the product of the sensitivity of the CCD and the transmittance of the Bayer filter for the corresponding wavelength), respectively. *ε_λ_* is the molar absorptivity (M^−1^·cm^−1^) of the Au(III), and *s_λ_* and *L* are the dispersion coefficient (cm^−1^) and the thickness of the solid support, respectively.

If the concentration of the absorbing species is zero (namely, before the reaction), Equation (3) can be written as follows:(7)$$R_{0}=AE_{0,R}$$

To avoid the effect of constant *A* on the analytical signal, the parameter *R_r_* (1) is defined as:(8)$$R_{r}=\frac{\left(R_{0}-R\right)}{R_{0}}=\frac{E_{1,R}}{R_{0}}c-\frac{E_{2,R}}{R_{0}}c^{2}$$

For parameters G and B, similar equations can be deduced.

Figure 4 shows a second-degree polynomial correlation between *R_r_* and tyramine concentration (*R_r_* changes lineally for low tyramine concentrations).

The limit of detection (LOD) was calculated by substituting in Equation (6) the signal corresponding to three times the standard deviation of the blank, resulting in a LOD value of 0.23 μM. The reproducibility for a tyramine concentration of 30 µM (*n* = 5) for each RGB coordinate is summarized in Table 2.

Table 2 shows the RSD of the measured RGB coordinates, indicating that the method produces semi-quantitative results or naked-eye detection capability. It is important to realize that these measurements were performed using a smartphone without a lighting box (see Section 2.4). This procedure resembles in situ measurements carried out in food packaging, indicating that semi-quantitative tyramine is affordable using this type of sensor.

One of the most interesting features of the tested sensing layers is the remarkable selectivity for tyramine detection in the presence of other BAs. Thus, the response of the Au(III)/tectomer optical nanosensors to histamine (2-(4-Imidazolyl)ethylamine), putrescine (1,4-Diaminobutane), and cadaverine (1,5-Diaminopentane) was studied as follows:(a)Single BA tests: Au(III)/tectomer sensor layers were exposed to solutions of each of the tested BAs in the absence of tyramine. The formation of AuNPs was not observed in any case. This result is in good agreement with previous work reporting no AuNP generation during the enzymatic oxidation of these interfering Bas [14].(b)Double BA tests: Au(III)/tectomer sensing layers were exposed to tyramine in the presence of each of the other BAs at different molar ratios. In all cases, the tyramine concentration was set at 10 µM. Table 3 shows the achieved *R_r_* values (*n* = 5).

In enzymatic colorimetric methods for tyramine determination, histamine is usually a very strong interference, which usually affects the tyramine signal for histamine/tyramine ratios of 1:1 and even lower [14]. However, as shown in Table 4, our method provided high tyramine selectivity against histamine.

### 3.3. Sensing Based in Reflectance Measurements

Alternatively, more accurate and precise responses for tyramine detection can be achieved from reflectance measurements. Thus, we alternatively propose here a sensing procedure by which reflectance measurements were collected by synchronous scanning, where the amount of light transmitted by the sample is collected by the fluorimeter at the same wavelengths at which the sample is excited (see Section 2.5).

The absorbance spectra obtained using this method showed a maximum at 550 nm; this corresponded to the AuNP plasmon band (Figure 5a).

The absorbance value at 550 nm was plotted as a function of tyramine concentration, resulting in the calibration curve for tyramine concentrations ranging from 0.048 to 10 μM, as shown in Figure 5b, which is typical of absorbance measurements performed in solid supports. LOD and RSD values as low as 0.048 μM and 4.2%, respectively, were achieved by this method.

At high tyramine concentrations (>10 µM) the 550 nm AuNP plasmon band was upshifted (Figure 6a), indicating that the AuNP size significantly increases when the sensor layer is exposed to high tyramine concentrations, which is in good agreement with the RGB results shown above. By plotting the logarithm of the absorbance at 700 nm versus the tyramine concentration, a second-degree polynomial correlation was obtained between 10 µM and 3.0 × 10^−2^ M of tyramine.

### 3.4. Recovery Assay

Finally, this methodology was tested for tyramine determination in a cheese sample. A Gouda sample was purchased from a local supermarket and submitted to the tyramine extraction following a procedure described in the Appendix A. No tyramine was found in this sample. A recovery study was then carried out. To do so, the cheese extract was spiked with a 5.0 × 10^−6^ M tyramine solution, which was analyzed by the reflectance method. Five replicate determinations were performed and a 4.7 × 10^−5^ M (RSD = 2.4%) average value was obtained, corresponding to a 94% recovery. This result validates the capability of the real application of this method for real samples.

### 3.5. Comparison with Other Colorimetric Methods

Table 5 summarizes the most recently published methods for the determination of tyramine. As can be seen, the sensitivity of the proposed method was similar to these. In addition, it has other advantages such as not requiring the use of enzymes, avoids the use of dyes, and easy oxidation is usually found with these compounds; more importantly, the formation of nanomaterials occurs in a single step unlike other more complicated methods that require multiple and complicated steps. This method is therefore useful for the determination of tyramine in food, allowing for the prevention of food health problems.

## 4. Conclusions

Novel Au(III)/tectomer-based optical nanosensors for tyramine detection were successfully demonstrated. Tectomer coatings on PLA act as an efficient matrix for Au(III) immobilization, enabling non-enzymatic Au(III) reduction and AuNP formation. Au(III)/tectomer sensor layers were used for the semi-quantitative determination (or naked-eye selective identification) of tyramine in the presence of other BAs, particularly histamine, by obtaining the RGB coordinates of the resulting colored AuNPs via a smartphone. Alternatively, tyramine quantification can be achieved by measuring the absorbance value of the sensor layer, leading to a LOD of 0.014 μM. This methodology, based on the optical properties of Au(III)/tectomer layers, shows promise for applications in food quality control, smart food packaging, and medical diagnostics. The versatile chemical functionalization capabilities of tectomer surface amino groups may potentially allow this methodology to be extended to the detection of a wide range of analytes.

## Figures and Tables

**Figure 1 sensors-23-02524-f001:**
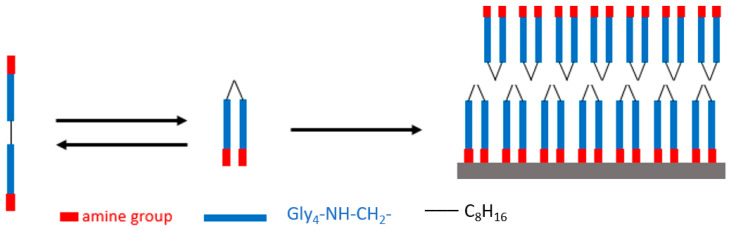
Assembly of biantennary oligoglycine into 2D structures, denoted as tectomers, stabilized by an extended hydrogen bonding network between amine groups.

**Figure 2 sensors-23-02524-f002:**
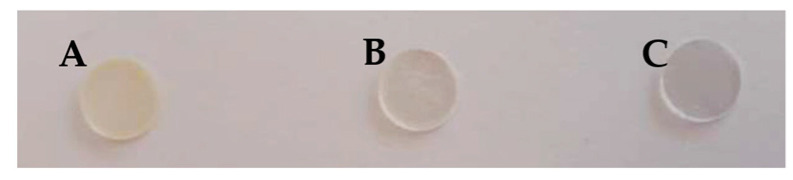
Au(III)/tectomer sensing layers on PLA supports prepared using a solution containing 3.0 × 10^−3^ M oligoglycine and different Au(III) concentrations: 3.0 × 10^−3^ M (**A**), 1.5 × 10^−3^ M (**B**), and without Au(III) (**C**) at pH 6.0.

**Figure 3 sensors-23-02524-f003:**
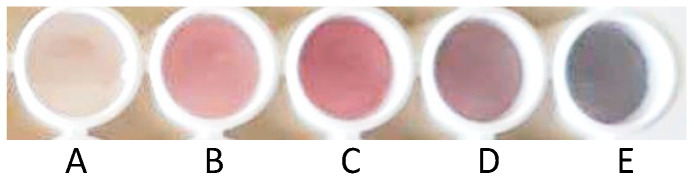
Au(III)/tectomer sensing layers on the PLA supports after exposure to different tyramine concentrations at pH 6.0: (**A**) 0 μM; (**B**) 3.0 μM; (**C**) 10 μM; (**D**) 30 μM; (**E**) 100 μM.

**Figure 4 sensors-23-02524-f004:**
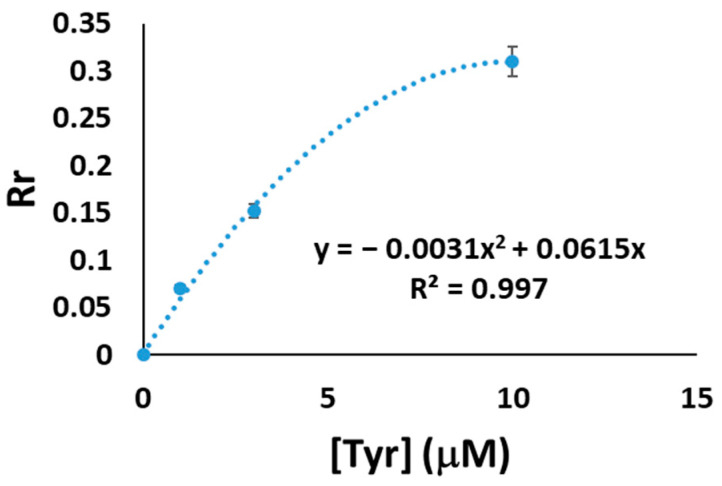
Calibration plot for the Au(III)/tectomer sensor layers fabricated in a 1:1 molar ratio in pH 6.0 phosphate buffer for tyramine detection.

**Figure 5 sensors-23-02524-f005:**
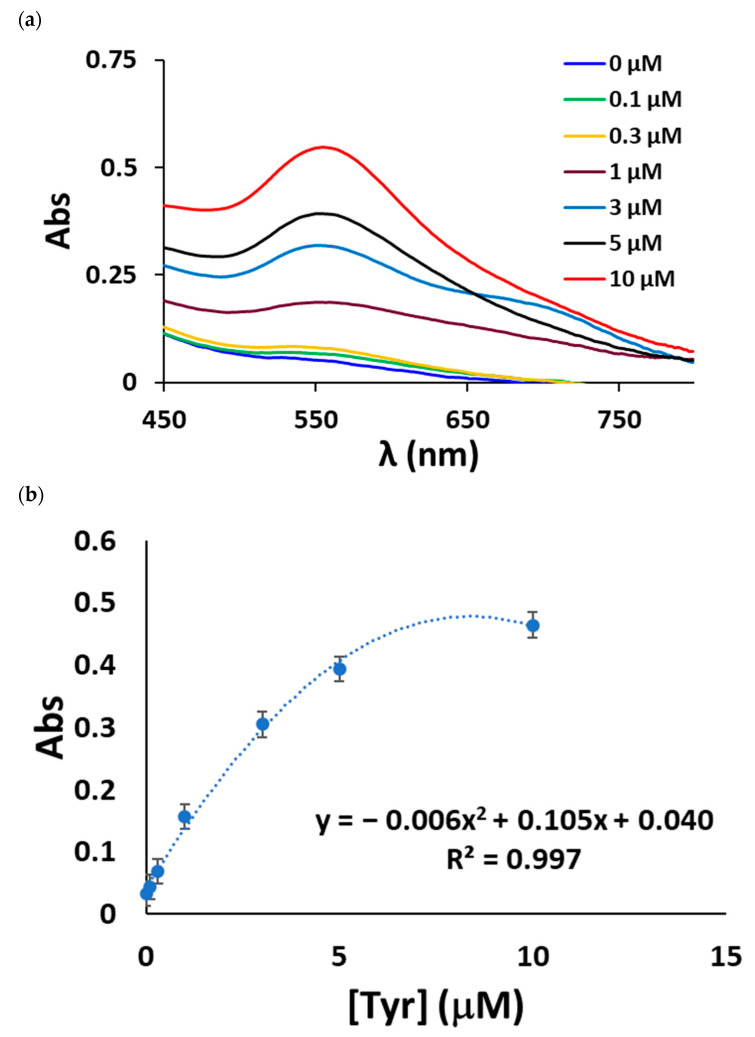
(**a**) Absorption spectra of the AuNPs collected from the reflectance measurements; (**b**) AuNP absorbance at 550 nm as a function of the tyramine concentration.

**Figure 6 sensors-23-02524-f006:**
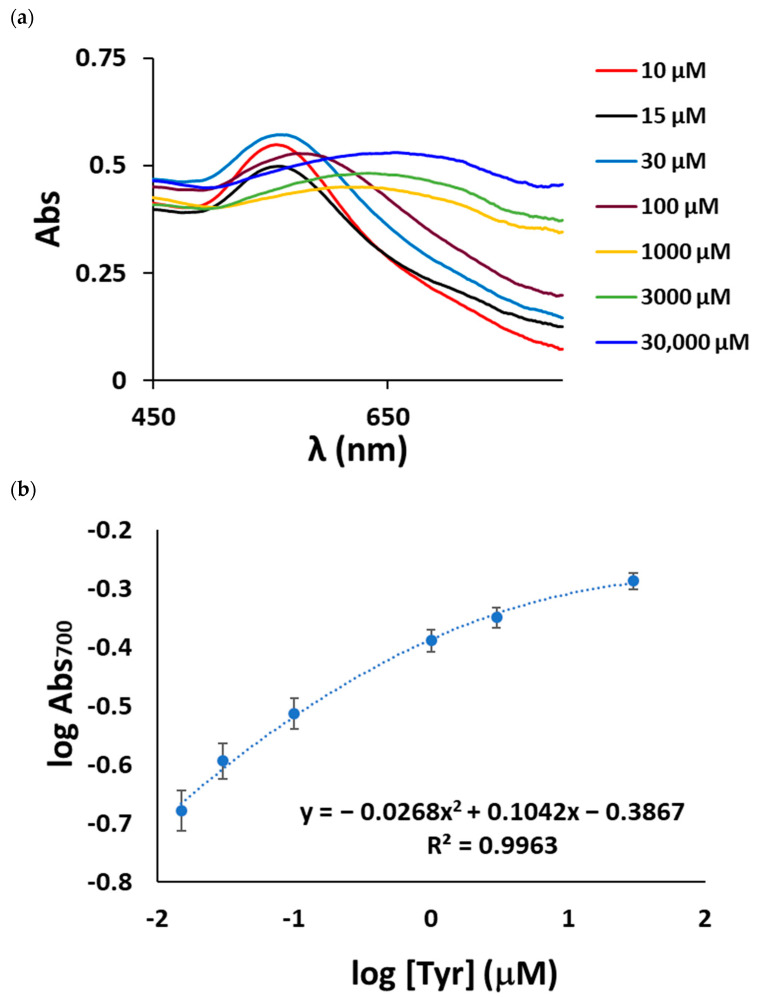
(**a**) The AuNP absorption spectra collected by reflectance measurements at high tyramine concentrations; (**b**) logAbs_700nm_ as a function of tyramine concentration.

**Table 1 sensors-23-02524-t001:** Tyramine concentration in different foods.

Food Product	Portion Size	Tyramine (mg)	Tyramine (µmol)
Canadian cheddar	28 g	43	314
Camembert	28 g	38	277
Bleu/Blue chesses	28 g	28	204
Gorgonzola	28 g	1.6	11
Cottage chesses, fresh	112 g	0	0
Tap beer	355 mL	38	277
Chicken livers, aged	28 g	60	438
Sauerkraut	112 g	3.5–14	25–102
Soy sauce	5 mL	0.05–4.7	3.6–34
Thai fish sauce	5 mL	0–3.7	27

**Table 2 sensors-23-02524-t002:** The RSD of the colorimetric determination of the 30 µM tyramine concentration from the RGB coordinates.

Coordinate	RSD (*n* = 5)
*R_r_*	23.81
*G_r_*	25.75
*B_r_*	31.17

**Table 3 sensors-23-02524-t003:** Response (*R_r_* values) of the Au(III)/tectomer sensing layers to tyramine in the presence of putrescine, cadaverine, or histamine at different molar ratios.

BA: Tyramine Ratio	Putrescine	Cadaverine	Histamine
0:1	0.310 ± 0.043	0.310 ± 0.043	0.310 ± 0.043
1:1	0.263 ± 0.037	0.316 ± 0.044	0.287 ± 0.040
2:1	0.310 ± 0.043	0.298 ± 0.042	0.298 ± 0.041
5:1	0.246 ± 0.034	0.316 ± 0.044	0.333 ± 0.047
10:1	0.269 ± 0.038	0.322 ± 0.045	0.263 ± 0.037
15:1	0.316 ± 0.044	0.322 ± 0.045	0.205 ± 0.029

**Table 4 sensors-23-02524-t004:** The response (*R_r_* values) of the Au(III)/tectomer sensing layers to the histamine/tyramine mixtures of high histamine:tyramine molar ratios.

Histamine:Tyramine Ratio	Histamine
0:1	0.310 ± 0.043
15:1	0.255 ± 0.029
20:1	0.211 ± 0.029
30:1	0.187 ± 0.026
40:1	0.135 ± 0.018
50:1	0.111 ± 0.016

**Table 5 sensors-23-02524-t005:** An overview on the recently reported optical methods for tyramine determination.

Composition of Sensor	Response Time	Analyte	Detected Signal	Anal. Range	LOD	Ref.
AgNPs (preliminary HPTLC separation of the analyte	-	Tyramine	Raman (λ_exc_ = 633 nm)	30–80 mg/kg	-	[29]
pH indicator dye and Remazol Brilliant blue immobilized on cellulose microplates	1.5 h	Total Biogenic Amines	CIE lab color space	0.3–30 mg/kg	-	[5]
Luminescence readout cellulose acetate nanofibers embedded with Py-1	20 min	Tyramine	Fluorescence with RGB/digital camera	1.37–13.7 mg/kg	0.4 mg/kg	[30]
Microliter plate with sensor film based on Py-1 embedded in Hypan HN80	10 min	Total Biogenic Amines	Fluorescence	0.5–70.0 mg/kg	0.165 mg/kg	[31]
Gra-QDs@MIPs	50 min	Tyramine	Fluorescence	0.07–12 mg/kg	0.02 mg/kg	[32]
Melanin-UCNPs NaGdF4:Yb/Er@NaYF4	45 min	Tyramine	Fluorescence	0.02–4.57 mg/kg	0.004 mg/kg	[33]
Fluorescent organic nanoparticles (FONs) with tetrapodal receptor	-	Tyramine	Fluorescence	27.4–219.5 mg/kg	0.05 mg/kg	[34]
AuNPs formation	≈30 min	Tyramine	Color generation	3.4–45.3 mg/kg	0.46 mg/kg	[14]
This work (RGB)	3 h	Tyramine	RGB with smartphone camera	0.107–1.37 mg/kg	0.0315 mg/kg	
This work (reflectance)	3 h	Tyramine	Reflectance measurements	6.57 µg/kg–1.37 mg/kg	1.97 µg/kg	

## Data Availability

Not applicable.

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
