# Peer review of "Tectomer-Mediated Optical Nanosensors for Tyramine Determination"

_sensors, 2023, doi:10.3390/s23052524_

Round 1

Reviewer 1 Report

In the manuscript, the authors described a new method to determine tyramine. There are some valuable scientific results in the study in terms of discovering or improving a new approach to a rapid determinion of tyramine. However, the study sounds as not completed since the method they describe based on color determinations, therefore, the food matrix or other matrixes which possibly contain tyramine are high likely to affect color determinations. In this study, the sensor was tried on the analysis of tyramine in poor solutions. The validation studies were not completed in the study. For example, the recovery of the method is unclear. Moreover, the tyramine values used in the study are very low. In the spoilage or fermented foods, tyramine values are expected very high. Therefore, the study seems to show very limited information on the performance of the developed sensor.

The authors explained that purpose of the study as the need of a rapid method for tyramine analysis due to food spoilage or safety. However, there is no regulations set by goverment authorities on tyramine although high contents present health risk for fermented food products. The authors should explain which amounts of tyramine in food products present the quality and safety issues. Moreover, the amounts selected to evaluate the method must relate to such levels. Low amounts are not usually the risk. So the method should also suitable to detect high levels.

Author Response

Thank you very much for your evaluation. In your comment, there are several different aspects to be considered.

1) The possible interference from the color of the food or another type of samples.

This will occur if the color of the matrix is very intense (more than that of the formed AuNP). Anyway, as the color of the matrix will not change during the sensing reaction, this color can be subtracted by taking a picture before the reaction.

2) Regarding tyramine concentration.

In the paper two calibration lines have been included: one for low tyramine concentrations (figure 4) and another one for high tyramine concentrations (figure 5). It is not sure that food samples containing even higher tyramine concentrations could be found. Anyway, the time necessary for obtaining the final color depends on the tyramine concentration to be measured. In the final edition of the paper we choose 180 min in to give time enough for the lowest tyramine concentration to react. If higher concentrations of tyramine need to be determined, the reaction could be modified accordingly. In addition, Au(III) concentrations in the tectomer could be fitted to the tyramine concentrations to be tested.

3) Regarding foods having tyramine concentration that currently need to be controlled.

They are mainly cheeses (“cheese syndrome”). A new table (Table 1) has been included in the new edition of the manuscript dealing with this information.

4) Finally, the paper in just a proof of concept.

When the system is going to be used for the tyramine control in specific foods, it should be necessary to perform additional studies regarding to measure the migration rate of tyramine from the food to the sensor and the distribution constant food/tectomer. This would need specific and extensive additional assays.

Reviewer 2 Report

Please refer to the enclosed file.

Reviewer 3 Report

This manuscript introduced a simple colorimetric sensor to detect tyramine. However, there would be several concerns with the submitted paper, as listed below.

1. The clarity of all pictures is not enough.

2. How specific is the detection of tyramine? How does the detection environment (e.g., temperature, pH etc.) affect the test results?

3. It is recommended to supplement and compare the detection limits (LOD) of tyramine by similar colorimetric methods.

4. It is suggested to supplement the latest relevant references.

5. Please refer to the format of the bibliography according to the requirements of the journal Biosensor.

Round 2

Reviewer 1 Report

The authors have improved the manusript. However, the validations test are required to be improved. The spelling of th text must be checked again. For example, in the first page which is the abstract: 'Relative Standart Desviation' must be changed to 'Relative Standart Deviation'.

Author Response

Thank you very much for your evaluation and suggestions.

In the new edition of the manuscript we have included a recovery study of tyramine in a Gouda cheese extract. A 94% recovery has been obtained.

Moreover, we have review for misspelled word throughout the paper and they have been corrected.

Reviewer 3 Report

Two questions should be solved before the manuscript accepted

1. I saw a paragraph in the introduction "However, there are individuals who are intolerant to tyramine and other BAs, either because their body is unable to product the corresponding amine oxidases, or because of enzyme inhibition due to drug ingestion. ", it seems that your approach should solve the above problem logically. But I don't see the uniqueness of the article's method from the experimental results or the discussion section. Please explain the uniqueness of your method from the Background, Experimental Results and Discussion section

2. The graphical representation of the results appears crude, and the icons do not clearly annotate the samples

Author Response

Q1. Thank you very much for your evaluation and suggestions.

According to your suggestions, the advantages of the proposed method have been highlighted throughout the manuscript (Introduction and Results sections).

Q2. Thank you for your comments. The quality of the figures in the manuscript has been improved. In particular, those figures containing spectra ( figures 4 and 5) have been submitted to smooth to improve clarity.

Round 3

Reviewer 3 Report

The authors did not take the reviewers' comments seriously, and there were no substantive changes in the Introduction, Figures, and Discussion.

Author Response

Dear Reviewer,

Thank you very much for your comments.

We are very sorry that you feel we have not taken seriously your comments. This was not our intention. Please find our explanation to your suggestions in the attached file.

Yours sincerely,

Susana de Marcos
